# Outcomes of a Self-Management Program for People with Non-Communicable Diseases in the Context of COVID-19

**DOI:** 10.3390/healthcare12161668

**Published:** 2024-08-21

**Authors:** Rodrigo Cesar León Hernández, Jorge Luis Arriaga Martínez, Martha Arely Hernández Del Angel, Isabel Peñarrieta de Córdova, Virginia Solís Solís, María Elena Velásquez Salinas

**Affiliations:** 1Consejo Nacional de Humanidades Ciencias y Tecnologías CONAHCYT, Mexico City 03940, Mexico; 2Facultad de Enfermería Tampico, Universidad Autónoma de Tamaulipas, Centro Universitario Tampico-Madero, Tampico 89339, Mexico; jorge.arriaga@uat.edu.mx (J.L.A.M.); marthahdz@docentes.uat.edu.mx (M.A.H.D.A.); pcordoba@docentes.uat.edu.mx (I.P.d.C.); 3Ministerio de Salud de Perú, Lima 15046, Peru; vsoliss@minsa.gob.pe; 4Dirección de Redes Integradas de Salud Lima Centro, Ministerio de Salud de Perú, Lima 15001, Peru; discapacidad@dirislimacentro.gob.pe

**Keywords:** depression, physical activity, self-management, non-communicable diseases

## Abstract

Objective: To evaluate the effectiveness of the online version of the Chronic Disease Self-Management Program (CDSMP) on physical activity and depressive symptoms in individuals with non-communicable diseases (NCDs) in Mexico and Peru during the COVID-19 pandemic. Materials and Methods: Quasi-experimental study with a non-probability sample of 114 people with NCDs, recruited by invitation in Mexico and by convenience in Peru. The participants were assigned to intervention (n = 85) and control (n = 29) groups. The Personal Health Questionnaire (PHQ-8) and the Physical Activity Scale were used to assess the outcomes. Measurements were taken before and after the intervention. The CDSMP comprises six sessions that take place once per week and last 2.5 h each. Results: The intervention group showed a significant reduction in depressive symptoms and an increase in physical activity (PA) at the end of the program. In contrast, the control group showed no significant improvement in depression and presented a significant decrease in PA.

## 1. Introduction

In the changing global health landscape, chronic or non-communicable diseases (NCDs) represent one of the greatest challenges of the 21st century. According to data from the World Health Organization (WHO), approximately 74% of global deaths are attributable to these diseases, which are primarily cardiovascular diseases, cancer, chronic respiratory diseases, and diabetes [1]. In addition to representing a direct threat to life, these diseases can also give rise to a range of psychological complications, with depression being a particularly prevalent consequence [2]. The epidemiological landscape in low- and middle-income countries is comparable. In Mexico, NCDs account for approximately 75% of deaths, with depression specifically affecting around 9% of the adult population. An increase in major depression cases has been observed among adolescents and young adults [3]. Similarly, these diseases pose a significant burden on public health in Peru, contributing to 73% of total deaths. Recent studies indicate that approximately 7% of adult Peruvians suffer from depression, with a notable rise in cases during and after the COVID-19 pandemic [4].

The interaction between NCDs and depression is “a two-way street” [5], i.e., while living with a chronic disease may increase the risk of developing depression, the incidence of depression also increases as these diseases accumulate [6]. The evidence suggests that 30% of people with an NCD suffer from depression, and this figure rises to 41% in the presence of multimorbidity, the presence of three or more diseases [7]. Likewise, depression can aggravate the symptoms of an NCD, hinder self-care, and worsen clinical outcomes [3,8,9]. It can also impede treatment follow-up and proper care of NCDs, increasing the risk of multiple complications [9] and their duration [5]. Furthermore, depression can ultimately impair an individual’s capacity to engage in daily activities and negatively impact their quality of life [9,10].

The management of NCDs encompasses three primary strategies: physiological support, risk-factor education, and regular physical activity [11]. In this sense, the scientific literature supports that regular physical activity (PA) is an effective treatment for these diseases [6,12,13,14]. However, a lack of PA also has detrimental effects on health; people who are physically inactive have an increased risk of mortality [13,15]. They are also twice as likely to experience depressive symptoms [16].

On the one hand, some studies have investigated the relationship between depression and PA, documenting that regular exercise not only benefits physical fitness but also plays a crucial role in improving the mental and emotional state of people living with NCDs [12,16]; one of the most notable findings is that adults who comply with regular PA guidelines have lower levels of depression [14,16,17,18]. The implementation of PA has been demonstrated to enhance a number of domains that are adversely affected by NCDs. These include body weight, fatigue, mental well-being, muscle and bone strength, physical capacity, and quality of life [11]. On the other hand, physical inactivity has been found to be associated with an increased risk of depression [16,19], particularly in women. [20].

Against this backdrop, there is a need for multimodal therapeutic approaches to address the problem of NCDs. Self-management programs, which aim to equip patients with skills and tools to manage their health, are a promising strategy. There is a large body of empirical evidence supporting the effectiveness of chronic disease self-management programs (CDSMPs or Tomando Control de su Salud (Spanish version)) across diverse formats, settings, and populations [21,22]. This research has evaluated outcomes that include improvements in quality of life [23,24,25], adherence [23,26,27], and reductions in hospitalizations and physician visits [24,28,29,30], which are promising for the improvement of depression and anxiety [21].

In Mexico, the CDSMP program has shown similar results in reducing depression and stress, improving social activity limitation, perception of quality of life, and communication with physicians. Concurrently, an augmentation in the level of physical activity, adherence to medical appointments, and self-management practice have been documented [31]. Finally, the CDSM program has demonstrated its effectiveness when implemented virtually. Participants in the program experienced significant improvements in self-efficacy, reduced pain and fatigue, and better chronic symptom management [32]. Additionally, there was a decrease in emergency room visits and hospitalizations [33]. During the COVID-19 pandemic, studies conducted in New York [34] and Washington [35] reported similar results with the virtual modality, including significant improvements in self-efficacy, a notable reduction in HbA1c levels and depression, and an increase in physical activity and healthy eating habits. In Mexico and Peru, enhancements in self-efficacy, disease knowledge, treatment adherence, and symptom management were also observed [36].

The literature presented above confirms the health problem represented by NCDs. It also provides evidence of the effectiveness of the CDSMP, particularly on the variables of PA and depression. However, there is a lack of studies on the impact of the program in the Latin American region, especially during the COVID-19 pandemic, which was a complicated historical moment for people living with such diseases.

In the context of the pandemic, people with NCDs were considered a vulnerable group, as they were more susceptible to infection or death due to the virus [37] and to developing severe complications from COVID-19 [38,39,40]. Furthermore, indirect effects of COVID-19 on the treatment and self-management of people with NCDs were found. For example, social distancing measures affected the attendance of visits to specialists and the lack of production of NCD drugs [41], as well as the effects of unemployment and lack of access to social security [42].

It is also known that the COVID-19 pandemic caused drastic changes in daily life, with confinement and social distancing limiting the usual physical activities [40]. These limitations created an environment in which maintaining adequate levels of PA could be challenging and highlighted the need to deliver programs remotely, safely, and efficiently [37,43]. Given this situation, it became imperative to adapt and implement virtual programs [44] that promote PA and a healthy mental state.

In view of the serious problem of NCDs and their impact on PA and mental health, which were exacerbated during the pandemic, as well as evidence of the positive results of the aforementioned program, the aim of the present study was to evaluate the effectiveness of the online version of the CDSMP program on physical activity and depressive symptoms in people with NCDs in Mexico and Peru during the COVID-19 pandemic.

## 2. Materials and Methods

### 2.1. Participants

The non-probabilistic initial sample consisted of 132 people with non-communicable diseases. The sampling was mixed: by invitation in Mexico and by convenience in Peru. Those with a diagnosis of NCD of more than 3 months at the time of the study were included. Those under 18 years of age and those who did not respond to all the instruments were excluded. The participants were consecutively assigned to intervention and control groups; the final sample consisted of 114 participants: intervention group (n = 85) and control group (n = 29). The process for forming the study sample is shown in Figure 1. The design of this study was quasi-experimental with independent (intervention and control) and repeated (pretest–posttest) measurements.

### 2.2. Measuring Instruments

The Personal Health Questionnaire (PHQ) was used to measure depression. The version validated in Mexico consists of 8 items and a Likert-type scale with 4 options (from 0 points = no day to 3 = almost every day). An overall score is obtained by summing the responses and is categorized into no depressive symptoms, 0–4 points; mild symptoms, 5–9; moderate symptoms, 10–14; severe symptoms, 15–19; and very severe symptoms, 20–24 points. The internal consistency was α = 0.78 [45]. Physical activity was measured using the Lorig et al. scale [46], which consists of 6 items assessing the frequency of exercise in minutes per week, with 5 response options ranging from 0 = none to 4 = more than 3 h. It is interpreted using the average score when the items are added together. Its validation in Mexico showed an internal consistency of α = 0.62 and a test–retest reliability = 0.72 [45].

### 2.3. Procedures

Initially, approval for the use of the CDSMP program was obtained from authorities at the Faculty of Nursing, Tampico-Universidad Autónoma de Tamaulipas, Mexico, and the National Institute of Neoplastic Diseases (INEN), Peru. Trained facilitators or leaders then recruited people with NCDs. In Mexico, a call for applications was disseminated through social networks during the second half of 2021. In Peru, the INEN worker leaders invited the users of the institute to participate in the study during the first semester of 2022. Assignment to the intervention and control groups was non-randomized. Before and after taking part in the program, the instruments were applied (pretest and posttest measurements), which were automated using the Google Forms platform (https://workspace.google.com/products/forms/). The posttest was applied at the end of the sixth session in the intervention groups and the same day in the controls.

The CDSMP program, an online format, consists of six sessions of 2.5 h each, meeting once a week, and is facilitated by two people who have been previously trained and certified by Stanford University. Its content addresses topics related to healthy eating, exercise, decision-making, symptoms such as sleep, pain, and fatigue, managing difficult emotions, problem-solving, communication skills, and goal setting [47].

### 2.4. Ethical Considerations

This study is part of a project funded by the Consejo Nacional de Humanidades Ciencias y Tecnologías, CF-2023-G-1394, entitled “Self-Management Strategies to Improve the Health of Persons with Chronic Illness and Family Caregivers. A Networked Work”. It was approved by the Research Ethics Committee of the Tampico Nursing School, Universidad Autónoma de Tamaulipas, registration number FET/CI/2023/002, and by the Institutional Research Ethics Committee of INEN, letter No. 149-2020-CIEI/INEN. The study adhered to the guidelines established in the Regulations of the General Health Law on Health Research in Mexico [48] and the Helsinki Declaration [49].

### 2.5. Statistical Analysis

Frequencies, percentages, and means and standard deviations were used to describe the characteristics of the study sample. The Kolmogorov–Smirnov test was used to determine the normality criteria of the variables, which showed significant results (*p* < 0.05): Nonparametric inferential tests were used for the above: the Mann–Whitney U and Wilcoxon tests. The analyses were performed using the Statistical Package for the Social Sciences (SPSS-25).

## 3. Results

The sociodemographic characteristics show that 83.3% of the participants were female. The mean age was 58.8, with a standard deviation of 11.5. The predominant nationality was Peruvian (81.6%), marital status was married (53.4%), and the most frequent NCD reported by participants was cancer (28%) (see Table 1).

The results of the comparison test between the intervention and control groups (Mann–Whitney U) show that there were no significant differences (*p* > 0.05) in the depression and PA variables in the pretest measure, therefore, the groups were equal before starting the intervention. However, in the posttest measure, significant differences were detected in both variables (*p* < 0.001) (see Table 2).

To achieve the research objective, the Wilcoxon signed-rank test was applied. As can be seen in Table 2, significant results (*p* < 0.05) were seen in the intervention group for the two research variables (see Table 3).

With respect to the control group, the results of the analysis show statistically significant differences in PA (*p* = 0.003) but not in depression (see Table 4).

In order to clarify the direction of the changes of interest in the intervention group and the control group, graphs were plotted using the value of the median, before (pretest) and after (posttest) the intervention. In Figure 2, it can be seen that the intervention group showed a significant improvement in reducing depression, while the control group worsened during the same period.

Figure 3 shows a marked improvement in the PA level of the intervention group after the program, increasing from 75 to 135 min of exercise, in contrast to the control group, which experienced a reduction in activity level in the same time frame.

## 4. Discussion

The objective of this study was to evaluate the efficacy of the online version of the CDSMP program on physical activity and depressive symptoms in people with NCDs in Mexico and Peru during the COVID-19 pandemic. In response to the above, the results indicate significant improvements in PA and a reduction in depressive symptoms in the intervention group at the end of the CDSMP program. In contrast, the control group showed negative results in both variables, which were significant in the case of PA. In other words, users with NCDs who did not participate in the program during the pandemic period significantly decreased their PA practice. These findings demonstrate the efficacy of the online CDSMP to improve these two key aspects of health in individuals with NCDs, which, on the one hand, are considered an effective treatment for these diseases [6,12,13,14,20], and on the other hand, interact in a reciprocal manner leading to a deterioration in health [5,6,8,9,10].

The improvement in PA and the decrease in depression reflect results similar to previous studies [23,24,25,34], which also documented benefits in quality of life and adherence to post-intervention treatment. However, these findings provide new evidence of the effects of the CDSMP program in Latin American contexts, where there is a paucity of studies documenting these effects. In this regard, only the study by Peñarrieta et al. [31], in which they applied the same program in a face-to-face format, with a sample of people with NCD in Mexico, reported similar results with benefits in PA and depression at 3 and 6 months, compared with the control group. It should be noted that no empirical evidence was detected that reports any negative effects of the CDSMP program on PA and depression, therefore, the results of this study are similar to the international literature.

In regard to the control group, a negative and significant effect on PA was discerned. This confirms the findings of some authors who found that during the social distancing and health restrictions adopted during the pandemic, PA could be affected or limited, underscoring the importance of implementing distance programs [37,43,50]. These contributed to reducing the complications to which people with NCD were exposed [37,38,39,40], such as the increased risk of mortality and depressive symptoms [13,15,16].

With regard to the depression variable, the contrast between the results observed in the intervention and control groups (see Figure 2) allows us to affirm that the CDSMP program can be an extremely important tool, since it has been documented that depression interferes with self-care and treatment follow-up, in addition to worsening clinical outcomes, quality of life, and affecting daily activities [5,8,9,10].

In addition, the favorable results of the CDSMP online format could have a positive impact on some of the problems that were accentuated during the pandemic, such as non-attendance at specialist consultations or lack of access to social security [41,42]. Therefore, it is suggested to continue disseminating the program and provide evidence to support its efficacy in different Latin American countries, as well as its benefits in face-to-face and virtual formats. It is also suggested that the CDSMP be proposed among the health-related public policies of Mexico and Peru since there is evidence supporting the program’s contribution to reducing or postponing negative consequences on the health of people with NCDs.

With reference to the limitations of this study, the drawbacks derived from the social distancing that occurred during the COVID-19 pandemic are highlighted, which made it difficult to form the study sample. Thus, the non-randomization in the assignment to the intervention and control groups was also a limitation; however, the equality test between groups in the pretest measure ensured its initial equivalence. Finally, the absence of follow-up measures is also an important limitation, so it is suggested to perform measurements at 6 and 12 months in subsequent studies.

## 5. Conclusions

The TCS program in an online format proved to be an effective tool for improving physical activity and reducing depression in people with NCDs in Mexico and Peru, particularly in the context of the pandemic, when access to face-to-face health facilities was limited. Therefore, for future research, it is suggested to replicate self-management programs in virtual formats, as well as to carry out medium- and long-term evaluations, to address public health challenges in developing countries.

In conclusion, the results of this work contribute to the body of evidence supporting the CDSPM program worldwide. The practical implications are that the online CDSPM represents a valuable tool to combat NCDs in Latin American countries such as Peru and Mexico, where epidemiological indicators position them as serious health problems and health systems lack the capacity to provide adequate face-to-face services.

## Figures and Tables

**Figure 1 healthcare-12-01668-f001:**
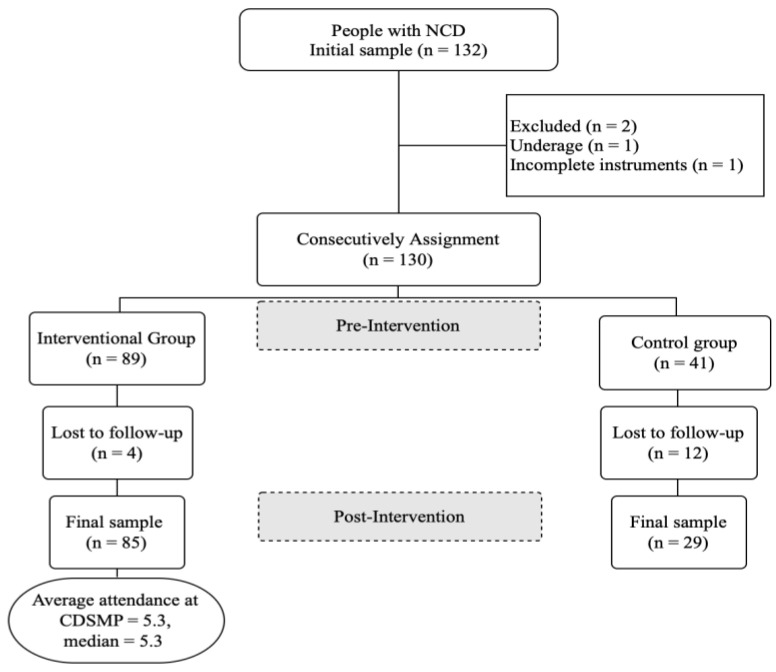
Study sample.

**Figure 2 healthcare-12-01668-f002:**
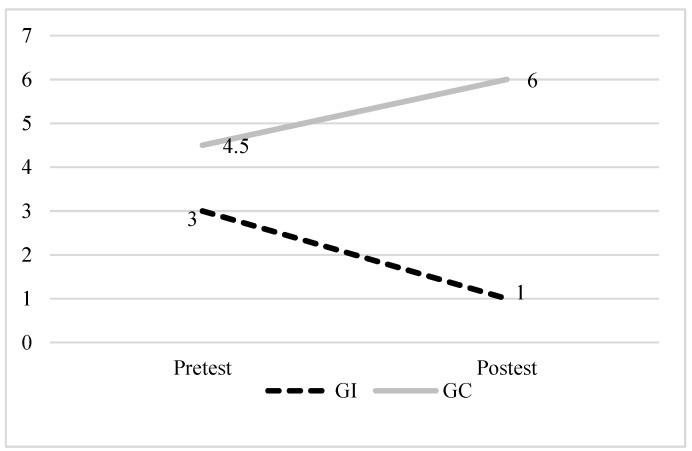
Results of the CDSMP program for the depression variable. Note: pretest and posttest medians for depression in the intervention (dashed line) and control groups (solid line).

**Figure 3 healthcare-12-01668-f003:**
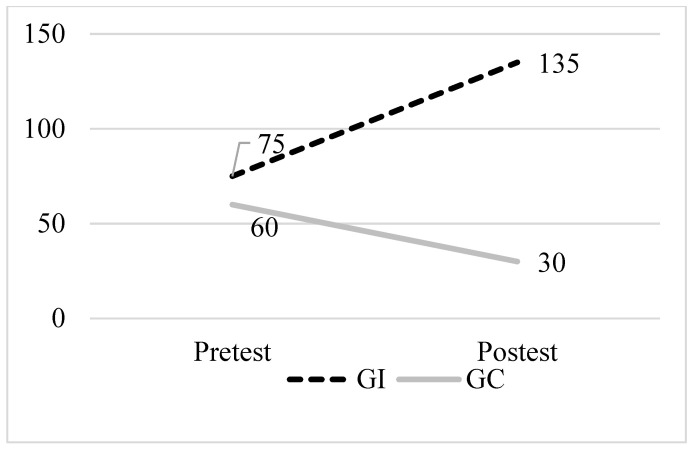
CDSMP program results for the physical activity variable. Note: pretest and posttest medians in the physical activity variable of the intervention (dashed line) and control groups (solid line).

**Table 1 healthcare-12-01668-t001:** Sociodemographic characteristics.

n = 114	*f*	%
Gender		
Female	95	83.3
Male	19	16.7
Nationality		
Perú	93	81.6
México	21	18.4
Marital status		
Married	61	53.4
Divorced	7	6.1
Single	27	23.6
Widower	19	16.9
First NCD reported		
Cancer	28	36.8
Diabetes	18	23.7
Hypertension	11	14.5
Other NCDs reported	19	25.0
Total =	114	100%

Note: ***f*** = frequencies, % = percentages.

**Table 2 healthcare-12-01668-t002:** Comparison between groups (intervention and control) in pretest–posttest measures.

Inter-Subjects	Intervention Groups (n = 85)			Control Groups (n = 29)			Sig.
	Mean Rank	Mean (SD)	Median	Mean Rank	Mean (SD)	Median	
Depression pretest	54.95	6.01 (5.0)	3	63.23	6.04 (4.9)	4.5	0.243
Physical activity pretest	57.50	120.4 (126.1)	75	57.52	133.4 (158.1)	60	0.973
Depression posttest	51.51	3.26 (4.1)	1	75.07	5.86 (4.3)	6	<0.001
Physical activity posttest	66.68	160.4 (124.2)	135	30.60	53.8 (62.9)	30	<0.001

Note: n = sample, Sig. = significance, SD = standard deviation.

**Table 3 healthcare-12-01668-t003:** Pretest–posttest depression and physical activity, intervention groups.

Intra-Subjects n = 85	Mean Rank		Mean (SD)		Median		Sig.
	Positive	Negative	Pretest	Postest	Pretest	Posttest	
Depression (R 0–24)	25.44	34.23	6.01 (5.0)	3.26 (4.1)	3	1	<0.001
Physical activity (R 0–1080)	39.21	43.50	120.4 (126.1)	160.4 (124.2)	75	135	0.006

Note: n = sample, Sig. = significance, SD = standard deviation, R = range.

**Table 4 healthcare-12-01668-t004:** Pretest-posttest depression and physical activity, control groups.

Pretest–Posttest n = 29	Mean Rank		Mean (SD)		Median		Sig.
	Positive	Negative	Pretest	Posttest	Pretest	Posttest	
Depression (R 0–24)	13.55	11.62	6.04 (4.9)	5.86 (4.3)	4.5	6	0.977
Physical activity (R 0–1080)	6.08	13.53	133.4 (158.1)	53.8 (62.9)	60	30	0.003

Note: n = sample, Sig. = significance, SD = standard deviation, R = range.

## Data Availability

The datasets used during the current study are available from the corresponding author upon reasonable request.

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
