# Peer review of "Outcomes of a Self-Management Program for People with Non-Communicable Diseases in the Context of COVID-19"

_healthcare, 2024, doi:10.3390/healthcare12161668_

Round 1

Reviewer 1 Report

Comments and Suggestions for Authors This is an interesting study, conducted virtually during a difficult period. The authors are to be congratulated. This article appear to report on the same study as reference 33 although different sets of outcome data are shown in each. A consort chart would be helpful as we are told that that those under 18 and those who did not respond to all the instruments were excluded. It is not clear if the total numbers before exclusion were 85 treatment and 29 controls or if these were the numbers after exclusion. There are no Ns given for the data tables. The timing of the post-test is not clear While the text says that the frequencies, percentages, means and standard deviations were used to describe the study sample but only frequency and percentage are given. I would like to see SD and means as well. For table 1 and table 2 I would like to see the actual means and SD of the data. I would like to see the range for each scale and the significance. Each table also need a Title describing the table. As is the tables, for this reviewer, give useful information other than the significance. Also the N= seem to be wrong as they are the same for both the treatment and control group. It is not clear why a statistical test was not done comparing the change scores of the treatment and control groups. I would like to see one table with the pre and post test scores (mean, median and SD) of both the treatment and the control groups and significance for both within each group as well as between the two groups. The two figures are interesting but all of these data could be put in one table. It would also be interesting to know the mean and median number of sessions attended. In some places, the tables are written in Spanish.

Comments on the Quality of English Language

N/A

Reviewer 2 Report

Comments and Suggestions for Authors

To strengthen the introduction, the authors could:

  • Include 1-2 more recent systematic reviews or meta-analyses on CDSMPs, if available.
  • Add a brief paragraph summarizing any previous studies on online/virtual CDSMPs specifically, as this study focuses on an online version.
  • Provide a bit more context on the situation for NCDs in Mexico and Peru specifically, as these are the study locations.

There are two tables labeled as Table 1. I would recommend reviewing this point and numbering the tables correctly.

In the discussion, I would like to know if there are previous studies that show differences with yours, and if they exist, to explore the possible reasons for these differences. On the other hand, were there any limitations in your study? What implications for the future might your study have? Could you delve into the possible differences observed between the two countries studied and what implications your study could have for public health policies in each country?

Reviewer 3 Report

Comments and Suggestions for Authors

Dear Authors,

The manuscript " Outcomes of a self-management program for people with non-communicable diseases in the context of COVID-19”," is an interesting document to the scientific community, that could be used to improve the quality of life of vulnerable populations.

The study presents coherence in its parts, from the introduction, objective, method, results, discussion and conclusions.

According to the objective: To evaluate the effectiveness of the online version of the Chronic Disease Self-24 Management Program on physical activity and depressive symptoms in people with non- communicable diseases in Mexico and Peru during the COVID-19 pandemic, the method is well described and approved by the ethics committee. Results are adequately described, and tables and graphs are clear. The statistical techniques are consistent with the results.

These conclusions are clear and are related to the objective and results found.

All the cited references are relevant to the research, 87% were published in the last 5 years.

Congratulations on your work.

Minor recommendations:

Clarify abstract; in line 30 where it is referred: “Measurements were taken before and after the intervention, which consisted of six weekly sessions of 2.5 hours each”, the clarification of what these sessions are consisted is recommended.

Review reference number 5.

Isabel Rabiais

Round 2

Reviewer 1 Report

Comments and Suggestions for Authors The authors have been very responsive to my comments and I think the article as is should be considered for publication.

Comments on the Quality of English Language n/a